# CaMKII-Dependent Contractile Dysfunction and Pro-Arrhythmic Activity in a Mouse Model of Obstructive Sleep Apnea

**DOI:** 10.3390/antiox12020315

**Published:** 2023-01-29

**Authors:** Philipp Hegner, Simon Lebek, Benedikt Schaner, Florian Ofner, Mathias Gugg, Lars Siegfried Maier, Michael Arzt, Stefan Wagner

**Affiliations:** 1Department of Internal Medicine II, University Hospital Regensburg, 93053 Regensburg, Germany; 2Department of Molecular Biology, University of Texas Southwestern Medical Center, Dallas, TX 75390, USA

**Keywords:** obstructive sleep apnea, oxidative stress, CaMKII, contractile dysfunction, arrhythmias

## Abstract

Left ventricular contractile dysfunction and arrhythmias frequently occur in patients with sleep-disordered breathing (SDB). The CaMKII-dependent dysregulation of cellular Ca homeostasis has recently been described in SDB patients, but these studies only partly explain the mechanism and are limited by the patients’ heterogeneity. Here, we analyzed contractile function and Ca homeostasis in a mouse model of obstructive sleep apnea (OSA) that is not limited by confounding comorbidities. OSA was induced by artificial tongue enlargement with polytetrafluorethylene (PTFE) injection into the tongue of wildtype mice and mice with a genetic ablation of the oxidative activation sites of CaMKII (MMVV knock-in). After eight weeks, cardiac function was assessed with echocardiography. Reactive oxygen species (ROS) and Ca transients were measured using confocal and epifluorescence microscopy, respectively. Wildtype PTFE mice exhibited an impaired ejection fraction, while MMVV PTFE mice were fully protected. As expected, isolated cardiomyocytes from PTFE mice showed increased ROS production. We further observed decreased levels of steady-state Ca transients, decreased levels of caffeine-induced Ca transients, and increased pro-arrhythmic activity (defined as deviations from the diastolic Ca baseline) only in wildtype but not in MMVV PTFE mice. In summary, in the absence of any comorbidities, OSA was associated with contractile dysfunction and pro-arrhythmic activity and the inhibition of the oxidative activation of CaMKII conveyed cardioprotection, which may have therapeutic implications.

## 1. Introduction

Sleep-disordered breathing (SDB) has emerged as a widespread disease with global burden, now affecting more than one billion patients worldwide [1]. In addition to SDB-related morbidity, these patients suffer from frequent co-morbidities such as hypertension [2], cardiac arrhythmias such as atrial fibrillation [3,4] with subsequent strokes [5], and heart failure [6,7]. Analysis from the *SchlaHF* registry, a large-scale multicenter study analyzing 6876 symptomatic patients with heart failure with reduced ejection fraction, detected moderate-to-severe SDB in 46% of the patients [7,8]. In heart failure patients above 80 years of age, the prevalence was even greater, at 59% [7,8].

Continuous positive airway pressure (CPAP) is currently the treatment option of choice for patients with SDB. However, CPAP therapy has recently failed to improve the long-term incidence of adverse cardiovascular events [9], which can partly be explained by the poor adherence to CPAP therapy, especially in patients with cardiovascular diseases [10]. Thus, novel therapeutic strategies to improve the cardiovascular outcome of patients with SDB are urgently warranted, which requires detailed knowledge of the cellular pathomechanisms.

Our group recently detected an increased activity of Ca/calmodulin-dependent protein kinase II (CaMKII) in patients with SDB, resulting in the pro-arrhythmic dysregulation of cellular Na and Ca homeostasis [11]. The dysregulation of cellular Ca handling has also been shown to impair electrical cell-to-cell communication via connexin-43 gap junction channels, further favoring the occurrence of cardiac arrhythmias [12,13,14,15]. Interestingly, the regulatory domain of CaMKII harbors two oxidation-sensitive methionine residues [16]. Upon oxidative stress, both methionine residues can become oxidized, resulting in increased CaMKII activation and further detrimental signaling, as we already demonstrated in atrial myocardium from patients with SDB [11,16,17]. However, since these patients exhibit different clinical characteristics (e.g., age, gender, etc.) and suffer from different comorbidities, it is very difficult to clearly assign a distinct pathomechanism solely to SDB. We therefore developed a new mouse model of obstructive sleep apnea (OSA) by injecting polytetrafluorethylene (PTFE) into the murine tongue [18]. This tongue enlargement resulted in an increased frequency of apnea and inspiratory flow limitations with subsequent hypoxia, recapitulating the key features of OSA in the absence of any potentially confounding comorbidity [18]. Importantly, PTFE-treated and control mice showed a similar breathing pattern during awake phases, indicating that there is no permanent upper airway obstruction [18].

In the present study, we hypothesized that PTFE-treated mice exhibit the specific CaMKII-dependent dysregulation of cardiac Ca homeostasis, resulting in contractile dysfunction and pro-arrhythmic activity. We further tested whether genetic ablation of the oxidative activation site of CaMKII (MMVV knock-in mouse) [19] or of the complete enzyme (CaMKIIδ knock-out mouse) [20] protects from these deleterious alterations.

## 2. Materials and Methods

All experiments involving mice are in accordance with the directive 2010/63/EU of the European Parliament, the Guide for the Care and Use of Laboratory Animals published by the US National Institutes of Health (NIH Publication No. 85–23, revised 1985), and with local institutional guidelines. The government of Unterfranken, Bavaria, Germany gave approval for the animal protocol (Protocol Number: 55.2-2532-2-512).

### 2.1. OSA-Induction by PTFE Injection

C57BL/6 wildtype, CaMKIIδ knock-out, and MMVV knock-in (genetic ablation of the oxidative activation site of CaMKIIδ) mice were randomly assigned to either the control (CTRL) group or to OSA-induction by PTFE injection (PTFE) (Figure 1). As previously described, PTFE (35 μm particle size; Sigma Aldrich, St. Louis, MO, USA) was injected into the tongue of male mice at an age of 8–12 weeks [18]. One hour before the PTFE injection, mice were treated with buprenorphine (0.1 mg/kg bodyweight intraperitoneal) for optimal analgesia. Anesthesia was established using intraperitoneal injections of medetomidine (0.5 mg/kg), midazolam (5 mg/kg), and fentanyl (0.05 mg/kg bodyweight). After that, mice were placed in a supine position on a heating plate. Body temperature was controlled by a rectal probe and anesthesia was continuously monitored by recording respiration and ECG. A total of 100 μL PTFE dilution (in 50% *w*/*v* glycerol, Sigma Aldrich) was injected into multiple sites of the base of the tongue using a 27-gauge cannula. Successful PTFE injection into the tongue was confirmed using the ultrasound technique (Vevo3100 system, VisualSonics, Toronto, ON, Canada). After that, anesthesia was antagonized using intraperitoneal injections of atipamezole (2.5 mg/kg), flumazenil (0.5 mg/kg), and buprenorphine (0.1 mg/kg bodyweight). All surgeries were performed by the same experienced investigator, who was blinded to the genotype of the mice.

### 2.2. Transthoracic Echocardiography

Cardiac function was assessed using transthoracic echocardiography with a Vevo3100 system (VisualSonics, Toronto, ON, Canada) and a 30 MHz center frequency transducer. Anesthesia was induced with 2% isoflurane (Isoflurane Vaporizer; VisualSonics, Toronto, ON, Canada) and maintained with 1.5% isoflurane under temperature, respiration, and ECG control. M-mode loops of the short- and long-axis views were recorded to determine the left ventricular ejection fraction. All measurements were performed by an experienced examiner blinded to the treatment (no PTFE vs. PTFE) and genotype of the mice.

### 2.3. Isolation of Ventricular Cardiomyocytes

Murine ventricular cardiomyocytes were isolated as previously described [21]. In brief, explanted hearts were mounted on a Langendorff perfusion apparatus and retrogradely perfused with (in mmol/L) 113 NaCl, 4.7 KCl, 0.6 KH_2_PO_4_, 0.6 Na_2_HPO_4_ × 2 H_2_O, 1.2 MgSO_4_ × 7 H_2_O, 12 NaHCO_3_, 10 KHCO_3_, 10 HEPES, 30 taurine, 10 2,3-butanedione monoxime, 5.5 glucose, and 0.032 phenol-red for 4 min at 37  °C (pH 7.4). After that, 7.5 mg/mL liberase TM (Roche diagnostics, Mannheim, Germany), trypsin 0.6%, and 0.125 mmol/L CaCl_2_ were added, and perfusion was continued until the heart became flaccid. Then, the murine ventricle was collected in perfusion buffer supplemented with 5% bovine calf serum. The tissue was cut into small pieces and disintegrated by pipetting until no solid tissue was left. Ca reintroduction was performed by increasing [Ca] stepwise from 0.1 to 1.0 mmol/L.

### 2.4. Measurements of Reactive Oxygen Species (ROS)

Isolated ventricular cardiomyocytes were plated on laminin-coated recording chambers and loaded with either 5 μmol/L CellRox^TM^ Orange (Thermo Fisher Scientific, Waltham, MA, USA) or 5 μmol/L MitoSox^TM^ Red (Thermo Fisher Scientific), both in the presence of 0.04% (*w*/*v*) pluronic acid (Invitrogen; 15 min incubation at 37 °C). Chambers were then placed on a laser-scanning confocal microscope (Zeiss LSM 700). Measurements were performed in normal Tyrode solution containing (in mmol/L) 140 NaCl, 4 KCl, 5 HEPES, 1 MgCl_2_, 10 glucose, and 1 CaCl_2_ (pH 7.4, room temperature with NaOH). Frame scans (CellRox^TM^ Orange: 555 nm excitation, LP 560 nm emission; MitoSox^TM^ Red: 488 nm excitation, LP 490 nm emission) were acquired every minute for 10 min upon electrical field stimulation (1 Hz). CellRox^TM^ Orange and MitoSox^TM^ Red fluorescence (F) was normalized to background fluorescence (F/F_0_). The slope of the increase in F/F_0_ over time was used as a measure of cellular (CellRox^TM^ Orange) and mitochondrial (MitoSox^TM^ Red) ROS production.

### 2.5. Epifluorescence Microscopy

Regular stimulated Ca transients were measured using epifluorescence microscopy. Therefore, isolated ventricular cardiomyocytes were plated on laminin-coated recording chambers and loaded with 5 µM of the Ca-sensitive dye FURA-2 AM for 15 min. The chambers were placed on the stage of an inverted microscope (Nikon Eclipse TE2000-U) and superfused with Tyrode solution containing (in mmol/L) 140 NaCl, 4 KCl, 5 HEPES, 1 MgCl_2_, 10 glucose, and 1 CaCl_2_ (pH 7.4, at 37 °C with NaOH). Regular Ca transients were elicited using electrical field stimulation (1 Hz, 20 V for 4 ms). Ca transients were obtained using a fluorescence detection system (IonOptix) and the Fura-2 fluorescence emission ratio was measured by alternating excitation at 340 nm and 380 nm. Steady-state Ca transients were analyzed using IonWizard software. Recordings were analyzed for the frequency of non-stimulated pro-arrhythmic events, defined as deviations from the diastolic Ca baseline between two stimulated transients. SR Ca leak was estimated by normalizing the Ca transient amplitude after 30 s of paused electrical stimulation to the amplitude before the pause, as previously described [11,22]. SR Ca content was measured by rapid caffeine application (10 mmol/L) and the quantification of the caffeine-induced Ca transient amplitude.

### 2.6. Statistical Analysis

Experiments were performed and analyzed in a way that was blind to the treatment (CTRL vs. PTFE) and genotype of the mice, and the results are presented as mean values per mouse ± standard error of the mean (SEM). The Shapiro–Wilk normality test was used to test for the normal distribution. A Student’s *t* test was used for the comparison of two continuous variables that were normally distributed, and the paired *t* test for paired data, as appropriate. One-way ANOVA with Holm–Sidak’s post hoc correction was performed for the comparison of more than two groups, and mixed-effects analysis with Holm–Sidak’s post hoc correction for paired data, respectively. GraphPad PRISM 9 was used to test for differences between linear regression analyses. Two-sided *p*-values below 0.05 were considered statistically significant.

## 3. Results

### 3.1. CaMKII-Dependent Contractile Dysfunction in OSA Mice

We characterized the cardiac function of control and OSA mice at baseline and 8 weeks after PTFE injection. Since CaMKII has previously been linked to contractile dysfunction, we also analyzed CaMKIIδ knock-out mice [22,23,24]. We recently demonstrated that ROS production is increased in patients with SDB, leading to increased levels of oxidized and thus overactivated CaMKII [11,17]. In order to test if the oxidative activation of CaMKII is required for CaMKII activation, the dysregulation of Ca homeostasis, and contractile dysfunction after PTFE treatment, we also included MMVV knock-in mice, where CaMKII is resistant to oxidative activation.

At baseline, there was no difference in the left ventricular ejection fraction between CaMKIIδ knock-in mice (MMVV), CaMKIIδ knock-out mice (CaMKII-KO), and wildtype (WT) littermates regardless of treatment assignment (CTRL/PTFE) (Figure 2A,B). Heart rate during echocardiography did not significantly differ between groups throughout (Figure 2E). Additionally, bodyweights were similar between groups (Figure 2F). Interestingly, in WT mice we observed a significant decrease in ejection fraction from 57.52 ± 0.95% at baseline to 51.42 ± 1.29% at 8 weeks after PTFE injection (*p* < 0.001, *n* = 25), which was also significantly lower compared with WT CTRL mice of equal age (left ventricular ejection fraction of 55.50 ± 0.72% at 8 weeks after study onset, *p* < 0.001, *n* = 9). Importantly, both MMVV (*n* = 6) and CaMKII-KO mice (*n* = 10) showed no impairment of cardiac function at 8 weeks after PTFE injection with ejection fractions of 56.42 ± 1.71% (*p* < 0.001) and 54.88 ± 1.19% (*p* < 0.001), respectively, compared with CTRL-treated mice (Figure 2A–C), but also compared with the corresponding baseline (Figure 2D). These data suggest that CaMKII oxidation may be required for CaMKII activation after PTFE treatment.

### 3.2. ROS Production Is Increased after PTFE Treatment

We previously demonstrated that ROS production is increased in patients with SDB [11]. However, patients differ in their clinical baseline characteristics (e.g., age, gender, body mass index, etc.) and may have different comorbidities (e.g., atrial fibrillation, diabetes mellitus, renal insufficiency), which may all have an impact on cellular ROS production, making it difficult to clearly assign this finding solely to SDB. Therefore, we measured cytosolic and mitochondrial ROS production in isolated ventricular cardiomyocytes from WT mice at 8 weeks after PTFE or CTRL treatment (Figure 3).

Linear regression analysis revealed a significant increase in CellRox^TM^ Orange fluorescence intensity as a function of time in the cytoplasm of cardiomyocytes isolated from PTFE-treated mice (*p* < 0.001, r^2^ = 0.839, *n* = 7), which was significantly different from CTRL-treated mice (*p* < 0.001, Figure 3A,B). This corresponded to a 2.3-fold increase in the CellRox^TM^ Orange fluorescence rate (in ΔF/F_0_*min^−1^) of 0.014 ± 0.001 in PTFE mice compared with 0.006 ± 0.001 in the control (*p* < 0.001, *n* = 7 each, Figure 3C).

In accordance, mitochondrial ROS production (measured as MitoSox^TM^ Red fluorescence intensity) increased over time in PTFE mice (*p* < 0.001, r^2^ = 0.656, *n* = 7) and was significantly greater compared with isolated myocytes from CTRL-treated mice (*p* < 0.001, *n* = 7, Figure 3D,E). The rate of increase in MitoSox^TM^ Red fluorescence intensity rose from (in ΔF/F_0_*min^−1^) 0.010 ± 0.002 in the control to 0.019 ± 0.002 in PTFE mice (*p* = 0.011, *n* = 7 each, Figure 3F).

### 3.3. CaMKII-Dependent Dysregulation of Cellular Ca Homeostasis in OSA Mice

Dysregulation of cellular Ca homeostasis is a key feature of contractile dysfunction and is critically regulated by CaMKII [22,23,24]. Therefore, we measured Ca transients in isolated ventricular cardiomyocytes at 8 weeks after PTFE injection using epifluorescence microscopy. We found Ca transient amplitude to be decreased from (in ΔF_340_/F_380_) 0.400 ± 0.028 in WT CTRL mice (*n* = 21) to 0.307 ± 0.018 in PTFE-treated animals (*p* = 0.007, *n* = 22, Figure 4A,B). Importantly, cardiomyocytes from MMVV knock-in mice treated with PTFE (*n* = 8) were fully protected and showed a normal Ca transient amplitude of (in ΔF_340_/F_380_) 0.441 ± 0.039, which was significantly higher than in WT mice after PTFE treatment (*p* = 0.007) and was comparable to MMVV CTRL mice. We did not observe differences in diastolic Ca levels or relaxation parameters (RT_70_ and RT_90_, Figure 4C–E).

To test if CaMKII activation after PTFE treatment would result in increased pro-arrhythmic activity, we analyzed non-stimulated pro-arrhythmic events that were defined as deviations from diastolic Ca baseline between two stimulated Ca transients. Interestingly, compared to CTRL, we observed a >2⎼fold increased frequency of pro-arrhythmic Ca release events in WT mice treated with PTFE (in s^−1^: 0.043 ± 0.005 vs. 0.021 ± 0.003 in WT CTRL, *p* < 0.001). In contrast, PTFE treatment did not result in increased pro-arrhythmic activity in cardiomyocytes isolated from MMVV mice (0.022 ± 0.003, *p* = 0.011 vs. WT PTFE, Figure 4F).

To estimate diastolic SR Ca leak, we paused electrical stimulation for 30 s and compared the Ca transient amplitude after the pause with that before the pause (Figure 5A). An increased rate of SR Ca leak would deplete the SR Ca storage, meaning that less Ca is available for the next stimulated transient [11,22]. Consistent with an increased rate of SR Ca leak, PTFE treatment resulted in a significant decrease in the Ca transient amplitude ratio after a 30 s pause in cardiomyocytes from WT mice (1.282 ± 0.043 vs. 1.420 ± 0.048 for WT CTRL, *p* = 0.032, Figure 5B). In contrast, cardiomyocytes isolated from MMVV mice after PTFE treatment exhibited a normal post-pause behavior (1.518 ± 0.068, *p* = 0.014 vs. WT PTFE).

The depletion of SR Ca content due to diastolic SR Ca leak may explain the observed systolic contractile dysfunction of WT mice after PTFE treatment. Therefore, we measured the Ca transient evoked by rapid caffeine application in isolated ventricular myocytes as a measure of SR Ca content. Consistent with the increased SR Ca leak, the caffeine-induced Ca transient amplitude was significantly decreased to 0.500 ± 0.046 (*n* = 13) in WT mice after PTFE treatment (*vs*. 0.747 ± 0.071 in WT CTRL mice, *p* = 0.007, *n* = 12, Figure 5C,D). In contrast, PTFE treatment did not reduce caffeine-induced Ca transient amplitude in MMVV mice (0.853 ± 0.035, *n* = 4), which was significantly greater than in WT PTFE (*p* = 0.007).

## 4. Discussion

In the present study, we applied a novel mouse model of obstructive sleep apnea and investigated left ventricular function, cardiomyocyte Ca handling and pro-arrhythmic activity. We found systolic ventricular function to be impaired, likely due to decreased Ca transients as a consequence of increased diastolic SR Ca leak, leading to the depletion of the SR Ca content. Importantly, genetic ablation of the oxidative activation sites of CaMKIIδ conveyed full cardioprotection from these OSA-induced pathogenic alterations. This suggests that OSA—via enhanced ROS production, which we also measured in the present study—results in CaMKII oxidation and activation, leading to disturbed Ca handling with contractile dysfunction and even pro-arrhythmic activity. This is the first study to investigate this novel oxidative pathway in an OSA mouse model devoid of the multiple confounding comorbidities usually found in patients with SDB.

### 4.1. Mechanisms of Cardiac Disease in SDB

In recent years, SDB has evolved into a disease with global socioeconomic relevance [1]. Amongst several other comorbidities, SDB is frequently associated with arrhythmias such as atrial fibrillation that substantially influence patients’ morbidity and mortality [1,3,4,25]. Interestingly, several mechanisms have already been independently shown to be at play, while the detailed interactions remain insufficiently understood [6,11,17,26,27]. Increased expression and activity of CaMKII has been shown to be a hallmark for arrhythmias and heart failure [22,23,28,29,30]. There are multiple features of SDB that have been reported to independently increase CaMKII activity, namely reactive oxygen species (ROS) following hypoxemia/reoxygenation [16,31,32], beta-adrenergic stress during sudden awakening [31,33], and atrial wall stress due to upper airway obstruction [31,34,35]. Indeed, we recently detected increased CaMKII activation in atrial biopsies of patients with SDB [11].

CaMKII has been shown to be subject to post-translational modifications that prevent the association of the autoinhibitory region with the catalytic domain, thus promoting increased CaMKII activity [16]. Several clinical characteristics of SDB are critically linked to systemic and myocardial ROS production, potentially favoring CaMKII activation through the oxidation of M281 and M282 [16,36,37,38]. Cyclic episodes of hypoxia/reoxygenation are a main inductor of oxidative stress (ROS) [38]. In patients with SDB, arousals from sleep dramatically increase the beta-adrenergic system, resulting in the activation of NADPH oxidase and subsequent ROS production [36,37,39]. Beta-adrenergic signaling has been shown to activate the renin angiotensin aldosterone system (RAAS), which is another strong inductor of ROS [16,36,37,39]. Especially in OSA, each forced inspiration against the obstructed airway during apnea results in a dramatic negative intrathoracic pressure with increased afterload [36,37,38,40]. This increases myocardial oxygen demand and may even further promote ROS generation [36,37,38,40].

In the present study, we analyzed the effect of OSA induction on cardiac function in a mouse model, where both oxidation-sensitive methionines are replaced with valines, thus rendering CaMKII insensitive to oxidative activation (MMVV knock-in) [19]. Importantly, MMVV knock-in conferred full cardioprotection against OSA-induced impairment of cardiac function, which was similar to CaMKII knock-out mice (Figure 2). We therefore conclude that the oxidation of M281 and M282 is the main driver of CaMKII activation in SDB. This conclusion is further supported by the increase in ROS production that we detected in both SDB mice (Figure 3) and previously in SDB patients [11].

### 4.2. CaMKII-Dependent Dysregulation of Cellular Ca Homeostasis

CaMKII has been shown to be a key regulator of cellular Ca homeostasis by phosphorylating ryanodine receptor type 2 (RyR2), Na/Ca-exchanger, phospholamban, and L-type Ca channel [23,41]. By phosphorylating RyR2 at S2814, CaMKII increases the diastolic open probability of RyR2, thereby increasing the diastolic SR Ca leak [23,28,41]. This results in the depletion of the SR Ca content with subsequent contractile dysfunction, as occurs in heart failure [22,23,24]. We previously linked SDB to increased CaMKII-dependent RyR2 phosphorylation with increased diastolic SR Ca leak and subsequent depletion of the SR Ca content [11,17,26]. In the present study, we can recapitulate this key feature of cardiac dysfunction in an OSA mouse model that is not limited by any confounding comorbidity. In particular, we found the Ca transient amplitude after 30 s of paused electrical stimulation to be decreased only in OSA wildtype mice, suggesting that the rate of SR Ca leak is increased [11,22]. Consequently, caffeine-induced Ca transient amplitude was decreased in OSA wildtype mice, indicating the depletion of the SR Ca content, which is in line with the impaired ejection fraction, as measured by echocardiography. Importantly, ablation of oxidative CaMKII activation (MMVV knock-in) protects from this deleterious Ca dysregulation in PTFE-treated mice.

Elevated cytosolic Ca levels following an increased SR Ca leak further increase the activity of CaMKII, possibly resulting in a detrimental vitious circle [42]. Moreover, the cytosolic Ca overload activates the Na/Ca-exchanger, which transports Ca to the extracellular compartment [43]. This is an electrogenic process, and the resulting net current favors the occurrence of delayed afterdepolarizations (DADs), which are important triggers of arrhythmias [11,22,28]. Indeed, we observed an increased frequency of pro-arrhythmic non-stimulated events in wildtype but not in MMVV knock-in OSA mice.

Besides the triggered pro-arrhythmic activity, the dysregulation of cellular Ca homeostasis has also been linked to the deterioration of electrical cell-to-cell communication, mediated by connexin-43 gap junction channels [12,13,14,15]. This may impair electrical conduction, further promoting the occurrence of cardiac arrhythmias [12,13,14,15]. Indeed, our group recently demonstrated that patients with SDB display a decreased atrial connexin-43 expression, which is also correlated with the occurrence of postoperative atrial fibrillation [27].

### 4.3. CaMKII Inhibition as a Potential Therapeutic Strategy in SDB

Investigating novel therapeutic strategies in SDB is highly relevant since, to date, CPAP is the treatment of choice for SDB-related diseases. However, as recently shown, CPAP intervention did not reduce the burden of paroxysmal atrial fibrillation in SDB patients [44] and patients’ compliance to CPAP therapy is low, especially in oligosymptomatic patients [45]. However, cardiovascular patients often do not experience excessive daytime sleepiness [46]. Moreover, adaptive servo-ventilation has even been shown to increase mortality in SDB patients with systolic heart failure and central apnea [47], and interventional therapies such as catheter-based ablation (e.g., pulmonary vein isolation) come with inherent risks [48]. Other therapeutic strategies correspond to the standard of care of non-SDB patients with heart failure (e.g., angiotensin-converting enzyme inhibitors, angiotensin receptor–neprilysin inhibitors, beta-blockers, mineralocorticoid receptor antagonists, gliflozins, etc.) or atrial fibrillation (e.g., beta-blockers, calcium channel antagonists, digitalis glycosides, amiodarone, anticoagulation, etc.) [49,50]. Unfortunately, even though novel therapeutic strategies for SDB are urgently warranted, the availability of SDB-specific therapy is very limited, wherefore detailed knowledge of the pathomechanisms is imperative.

Previously, we proposed CaMKII inhibition as an antiarrhythmic approach for patients with SDB [11]. Although there are already several CaMKII inhibitors under preclinical consideration, there is still no clinically available substance [22,51,52]. This circumstance may be explained by the challenge of substrate and CaMKII isoform specificity, which are required for a CaMKII inhibitor, but also by previous limitations regarding the bioavailability of the compounds [30,53]. However, there has been a great deal of effort to overcome these challenges in recent years. Recently, our group found the ability of ATP-competitive CaMKII inhibitor GS-680 to block diastolic SR Ca leak and to prevent multicellular arrhythmias in the atrial tissue of patients undergoing cardiac surgery. Moreover, treatment with GS-680 improved cellular Ca homeostasis in cardiomyocytes from patients with end-stage heart failure, showing that CaMKII inhibition is also a promising therapeutic concept in human ventricular cardiac tissue [22]. RA306 and RA608, two other ATP-competitive CaMKII inhibitors, have been shown to ameliorate cardiac function in vivo in a mouse model of dilated cardiomyopathy and in afterload-induced heart failure, respectively, overcoming the limited bioavailability of previous CaMKII inhibitors [54,55].

### 4.4. Study Limitations

We previously established PTFE injection into the tongue as a mouse model of OSA and showed that PTFE treatment increases the frequency of apnea and inspiratory flow limitations, with subsequent hypoxia [18]. Even though we refrained from re-validating the model in this study, future work is needed to link our findings on contractile dysfunction and Ca dysregulation to specific OSA parameters.

The aim of this study was to decipher a pathomechanism in the absence of any confounders. Therefore, we only analyzed male mice. However, there might be sex-dependent differences that need to be further investigated [6,56].

## 5. Conclusions

By analyzing an OSA mouse model without any potentially confounding comorbidities, we found that cardiac ROS production is increased, resulting in a dysregulation of cellular Ca homeostasis with pro-arrhythmic activity, depletion of the SR Ca content, and subsequent contractile dysfunction in vivo. Importantly, genetic ablation of the oxidative activation sites of CaMKIIδ conferred full cardioprotection from all of these deleterious events. Therapeutic strategies aiming to prevent CaMKIIδ oxidation may prove to be superior in improving contractility and preventing arrhythmias in patients with SDB. Indeed, a recent study demonstrated that CRISPR-Cas9 gene editing is able to specifically rewrite the *CaMKIIδ* gene in adult mice and to exchange both oxidation-sensitive methionines 281 and 282 with valines, which corresponds to the MMVV knock-in mouse model that we used in our present study [57]. According to the findings of our study, selective *CaMKIIδ* gene editing may be both applicable and beneficial for SDB, which will be investigated in the future. Moreover, detailed analysis of in vivo arrhythmias via implantable transmitters may be an interesting topic for a further study. Furthermore, the differential metabolism of cardiac energy substrates has been shown to be involved in both cardiac dysfunction and SDB [58,59,60]. Future studies will aim to investigate this mechanism using our SDB mouse model.

## Figures and Tables

**Figure 1 antioxidants-12-00315-f001:**
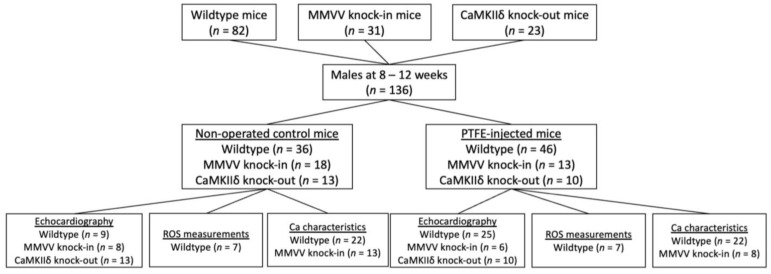
Study flowchart. C57BL/6 wildtype, MMVV knock-in, and CaMKIIδ knock-out mice were randomly assigned to either control or OSA induction by PTFE injection. After 8 weeks, echocardiography, ROS measurements, and epifluorescence microscopy were performed, whereby some experiments were performed in the same subject.

**Figure 2 antioxidants-12-00315-f002:**
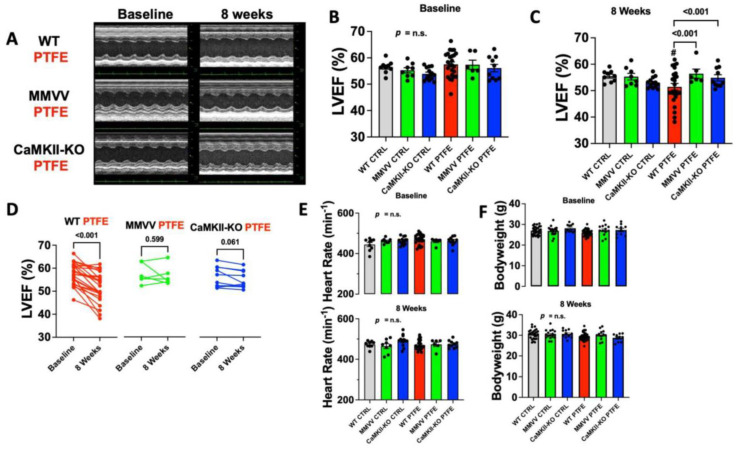
(**A**) Original M-mode echocardiographic traces of left ventricle. (**B**) Mean left ventricular ejection fraction per animal at baseline (CTRL groups: *n* = 9 WT, *n* = 8 MMVV, *n* = 13 CaMKII-KO; PTFE groups: *n* = 25 WT, *n* = 6 MMVV, *n* = 10 CaMKII-KO; ANOVA *p* = n.s.). (**C**) Mean left ventricular ejection fraction per animal after 8 weeks, # indicates *p* < 0.05 vs. WT PTFE at baseline. (**D**) Before–after presentation of PTFE group data from (**B**,**C**). (**E**) Heart rate during echocardiography from (**B**,**C**). (**F**) Animal body weight at baseline and after 8 weeks. Statistical comparisons are based on ANOVA, mixed-effects analysis with Holm–Sidak’s post hoc correction, and paired *t*-test, as appropriate.

**Figure 3 antioxidants-12-00315-f003:**
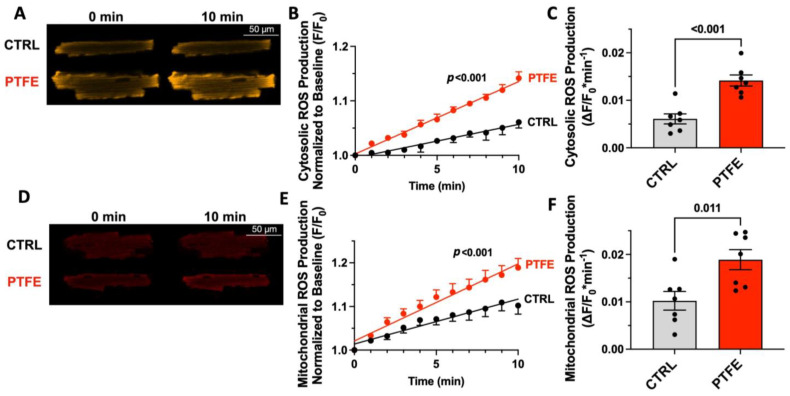
(**A**) Original confocal frame scans of isolated ventricular cardiomyocytes loaded with the cytosolic ROS fluorescence dye CellRox^TM^ Orange. (**B**) Fluorescence intensity over time, normalized to baseline (*p* < 0.001 for difference in linear regression slopes). (**C**) Mean data per animal for the increase in CellRox^TM^ Orange fluorescence intensity normalized to baseline fluorescence (ΔF/F_0_) at 10 min after start of recording (*n* = 7 vs. 7). (**D**) Original confocal frame scans of isolated ventricular cardiomyocytes loaded with the mitochondrial ROS fluorescence dye MitoSox^TM^ Red. (**E**) Fluorescence intensity over time, normalized to baseline (*p* < 0.001 for difference in linear regression slopes). (**F**) Mean data per animal for the increase in MitoSox^TM^ Red fluorescence intensity normalized to baseline fluorescence (ΔF/F_0_) at 10 min after start of recording (*n* = 7 vs. 7). Statistical comparisons are based on Student’s *t*-test.

**Figure 4 antioxidants-12-00315-f004:**
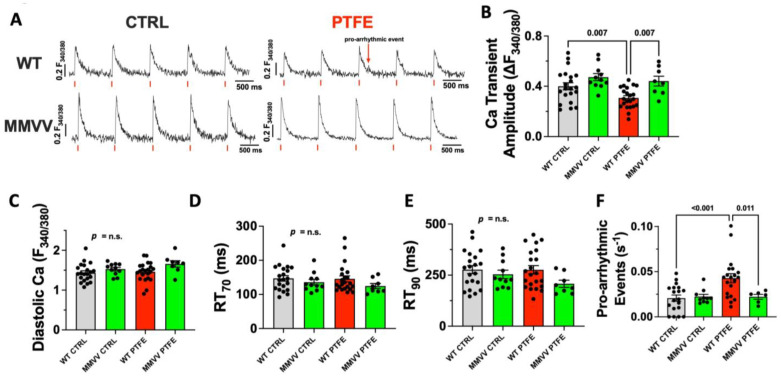
(**A**) Original Ca transients recorded in Fura-2 AM loaded isolated ventricular cardiomyocytes (1 Hz). The red arrow indicates a non-stimulated pro-arrhythmic event in a WT PTFE cell. (**B**) Mean Ca transient amplitude, diastolic Ca (**C**), relaxation time to 70% (**D**) and 90% of baseline (**E**) per animal (*n* = 21 WT CTRL, *n* = 11 MMVV CTRL, *n* = 22 WT PTFE, *n* = 8 MMVV PTFE). (**F**) Frequency of non-stimulated pro-arrhythmic events (as in (**A**), *n* = 19 WT CTRL, *n* = 10 MMVV CTRL, *n* = 20 WT PTFE, *n* = 6 MMVV PTFE). Statistical comparisons are based on ANOVA with Holm–Sidak’s post hoc correction.

**Figure 5 antioxidants-12-00315-f005:**
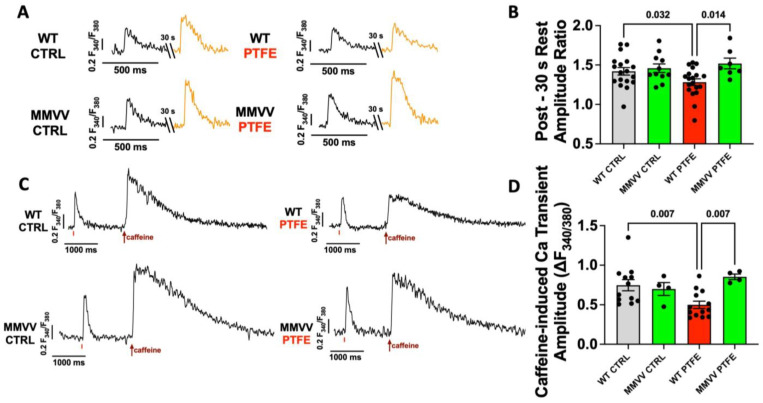
(**A**) Original Ca transients recorded in Fura-2 AM loaded isolated ventricular cardiomyocytes before and after 30 s cessation of electrical stimulation (post-rest test). (**B**) Mean Ca transient amplitude ratio per animal after the pause/before the pause as a measure of diastolic SR Ca leakage (*n* = 18 WT CTRL, *n* = 11 MMVV CTRL, *n* = 19 WT PTFE, *n* = 7 MMVV PTFE). (**C**) Original recordings of Ca transients induced by rapid caffeine application (red arrows), mean data per animal shown in (**D**) (*n* = 12 WT CTRL, *n* = 4 MMVV CTRL, *n* = 13 WT PTFE, *n* = 4 MMVV PTFE). Statistical comparisons are based on ANOVA with Holm–Sidak’s post hoc correction.

## Data Availability

Data is contained within the article.

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
