# Peer review of "CaMKII-Dependent Contractile Dysfunction and Pro-Arrhythmic Activity in a Mouse Model of Obstructive Sleep Apnea"

_antioxidants, 2023, doi:10.3390/antiox12020315_

Round 1

Reviewer 1 Report

In the paper “CaMKII-dependent contractile dysfunction and pro-arrhythmic activity in a mouse model of obstructive sleep apnea authors have applied a novel mouse model for obstructive sleep apnea and investigated left ventricular function, cardiomyocyte Ca handling and pro-arrhythmic activity. Authors found systolic ventricular function to be impaired likely due to decreased Ca transients as a consequence of increased diastolic SR Ca leak leading to depletion of the SR Ca content. The paper is well written but there are some points for improvement.

1)      Provide a flow chart of experimental design to clarify the groups and knock-out models.

2)      Did author measure the differential metabolism of the fuel molecules for heart in different models?

3)      List the potential therapeutics of Sleep-disordered breathing. Please include recent references of the medication.

Author Response

Reviewer 1:

In the paper “CaMKII-dependent contractile dysfunction and pro-arrhythmic activity in a mouse model of obstructive sleep apnea authors have applied a novel mouse model for obstructive sleep apnea and investigated left ventricular function, cardiomyocyte Ca handling and pro-arrhythmic activity. Authors found systolic ventricular function to be impaired likely due to decreased Ca transients as a consequence of increased diastolic SR Ca leak leading to depletion of the SR Ca content. The paper is well written but there are some points for improvement.

  1. Provide a flow chart of experimental design to clarify the groups and knock-out models.

Response: We thank the Reviewer for this comment and now provide a flow chart of the experimental design (novel Figure 1).

  1. Did author measure the differential metabolism of the fuel molecules for heart in different models?

Response: We thank the Reviewer for the effort and for this interesting idea. For the current manuscript, we did not measure the differential metabolism of the fuel molecules as we aimed to delineate the link between oxidative stress, CaMKII, and dysregulation of cellular Ca homeostasis in a novel mouse model of obstructive sleep apnea. However, we appreciate the Reviewer’s input and think it is a great idea for future work.

We have therefore added this perspective to the Conclusions section of the revised manuscript (page 11, lines 400-403):

Besides that, differential metabolism of cardiac energy substrates has been shown to be involved in both cardiac dysfunction and SDB [58-60]. Future studies will aim to investigate this mechanism using our SDB mouse model.”.

  1. List the potential therapeutics of Sleep-disordered breathing. Please include recent references of the medication.

Response: We thank the Reviewer for this important comment. In the revised version of the manuscript, we now include a section discussing current therapies for SDB and for SDB-related comorbidities (e.g., arrhythmias and heart failure). We have also included important recent references (e.g., the clinical guidelines from the European Society of Cardiology).

On page 10 (lines 346-361) it reads:

Investigating novel therapeutic strategies in SDB is highly relevant since to date CPAP would be the treatment of choice for SDB-related diseases. However, as recently shown, CPAP intervention did not reduce the burden of paroxysmal atrial fibrillation in SDB patients [44] and patients’ compliance to CPAP therapy is low, especially in oligosymptomatic patients [45]. However, cardiovascular patients often do not experience excessive daytime sleepiness [46]. Moreover, adaptive servo-ventilation has even been shown to increase mortality in SDB patients with systolic heart failure and central apnea [47], and interventional therapies such as catheter-based ablation (e.g., pulmonary vein isolation) come with inherent risks [48]. Other therapeutic strategies correspond to the standard of care of non-SDB patients with heart failure (e.g., angiotensin-converting enzyme inhibitors, angiotensin receptor-neprilysin inhibitors, beta-blockers, mineralocorticoid receptor antagonists, gliflozins, etc.) or atrial fibrillation (e.g., beta-blockers, calcium channel antagonists, digitalis glycosides, amiodarone, and also anticoagulation, etc.) [49,50]. Unfortunately, even though novel therapeutic strategies for SDB are urgently warranted, the availability of SDB-specific therapy is very limited, wherefore detailed knowledge of the pathomechanisms is imperative.”.

Reviewer 2 Report

The study by Hegner et al. investigated contractile function and Ca homeostasis in a mouse model of obstructive sleep apnoea.  Wildtype mice was compared to mice with a genetic ablation of the oxidative activation
sites of CaMKII. Sleep apnoea was induced by artificial tongue enlargement. The authors found in an OSA mouse an increased cardiac ROS production which induced a dysregulation of cellular Ca homeostasis with pro-arrhythmic activity, depletion of the SR Ca content, and contractile dysfunction. Genetic ablation of the oxidative activation sites of CaMKIIδ protected from these changes.

Comments: Were there any changes in the weight of animals, their behaviour, sleep apnoea or any other physiological parameter?

2.       Was sleep apnoea score similar in all groups?

3.       The authors measured ejection fraction of the heart. Heart rate, ECG analysis, arrhythmic episodes and other results obtained from the echocardiography should be also presented, for example in a Table.

Author Response

Reviewer 2:

The study by Hegner et al. investigated contractile function and Ca homeostasis in a mouse model of obstructive sleep apnoea. Wildtype mice was compared to mice with a genetic ablation of the oxidative activation sites of CaMKII. Sleep apnoea was induced by artificial tongue enlargement. The authors found in an OSA mouse an increased cardiac ROS production which induced a dysregulation of cellular Ca homeostasis with pro-arrhythmic activity, depletion of the SR Ca content, and contractile dysfunction. Genetic ablation of the oxidative activation sites of CaMKIIδ protected from these changes.

Comments:

  1. Were there any changes in the weight of animals, their behaviour, sleep apnoea or any other physiological parameter?

Response: We thank the Reviewer for this important comment. In the revised version of the manuscript, we now include the body weight at baseline and after 8 weeks (novel figure 2F). Animal body weights at 8 weeks were not significantly different between groups. Multiple comparisons of groups revealed that animal body weight at baseline was slightly lower in the WT PTFE group compared to the CaMKII-KO CTRL group (p=0.041), however due to the very minor mean difference of only 2.1 grams and no difference after 8 weeks (see above), we do not believe this to be of relevance. Animal behavior was normal throughout the study period. In addition to body weight, other physiological parameters such as heart rates were also similar between groups (see response for comments 2 and 3 below).

We have added this new data to the revised manuscript (page 5, line 180):

(…) bodyweights were similar between groups (Figure 2F).”.

  1. Was sleep apnoea score similar in all groups?

Response: We thank the Reviewer for this comment. We have previously validated the model in detail and found an increased frequency of apneas and inspiratory flow limitations with subsequent hypoxia (Lebek et al., PLoS One 2020). To reduce the total stress for the animals (which is strongly encouraged by the governmental regulators in Germany), we refrained from re-validating our previously published work. Nevertheless, we had performed a few exploratory confirmatory whole-body plethysmography experiments using few MMVV knock-in and CaMKII knock-out mice after PTFE treatment. These experiments indicated a similar inspiratory flow limitation after PTFE compared to WT mice treated with PTFE (data not shown).

  1. The authors measured ejection fraction of the heart. Heart rate, ECG analysis, arrhythmic episodes and other results obtained from the echocardiography should be also presented, for example in a Table.

Response: We thank the Reviewer for this comment. During echocardiography, animal heart rate was continuously monitored. We now present the heart rate during the echo for all groups at baseline and after 8 weeks (novel figure 2E). There were no significant differences in heart rate between groups (ANOVA p=n.s.). Even though analysis of ECG recordings acquired during echocardiography is not suitable in mice, we did not observe any overt arrhythmias during the echocardiographic measurements. However, for more detailed analysis of arrhythmias, continuous monitoring via surgically implantable telemetric transmitters might be feasible, which is an interesting idea and could be pursued in a new study.

Therefore, we have added above data and this perspective to the revised version of the manuscript (page 5, lines 178-180):

“Heart rate during echocardiography did not significantly differ between groups throughout (Figure 2E).”.

And in the Conclusions section (page 11, lines 399-400):

“Moreover, detailed analysis of in vivo arrhythmias via implantable transmitters may be an interesting topic for a further study.”.

Reviewer 3 Report

It is appreciated that the authors elucidated mechanisms that may underlie high pro-arrhythmic risk of individuals suffering from obstructive sleep apnoea, the clinically relevant and yet unresolved issue.  The study highlights the implication of CaMKII-dependent contractile dysfunction and pro-arrhythmic activity in a mouse model of obstructive sleep apnoea (OSA). Impact of CaMKII was explored using mice with a genetic ablation of the oxidative activation sites of CaMKII, which protect the mice against OSA induced impairment of heart function (ejection fraction). Isolated cardiomyocytes of OSA mice exhibited an increase of reactive oxygen species, decrease Ca2+ transients but increase of pro-arrhythmic activity. Nevertheless, some questions and remarks arise that should be addressed prior publication.

Abstract: What authors consider as pro-arrhythmic activity should be defined and included.

Introduction and Discussion: It is generally accepted that Ca2+ handling disorders resulting in Ca2+ overload and Ca2+ leaks form sarcoplasmic reticulum may trigger ventricular premature beats. However, it should be taken into consideration that Ca2+ disorders and Ca2+ overload deteriorate cell-to-cell electrical ensured by connexin-43 gap junction channels (Tribulova et al. 2009, Andelova et al. 2021). Such conditions result in impairment and block of conduction promoting occurrence of malignant cardiac arrhythmias. Attention should be payed to this fact.

Moreover, pro-arrhytmic changes or signalling discovered in the study should be defined more specifically as well as discussed.

 What is your explanation in support of pro-arrhythmic activity of cardiomyocytes, which exhibited decreased Ca2+ transients resulting in depressed contraction? Since just opposite, increased transients and diastolic Ca2+ concentration is considered as proarrhythmic.

How is SERCA activity in your OSA model? Reduced SERCA activity followed by reduced Ca2+ uptake by sarcoplasmic reticulum and Ca2+ overload may contribute to the contractility depression.

Concern the discussion, maybe less in more when focusing on key findings relevant to the aim of the study.

It would be fine to include perspectives in the context of the future research in chapter Conclusions.

Author Response

Reviewer 3:

It is appreciated that the authors elucidated mechanisms that may underlie high pro-arrhythmic risk of individuals suffering from obstructive sleep apnoea, the clinically relevant and yet unresolved issue. The study highlights the implication of CaMKII-dependent contractile dysfunction and pro-arrhythmic activity in a mouse model of obstructive sleep apnoea (OSA). Impact of CaMKII was explored using mice with a genetic ablation of the oxidative activation sites of CaMKII, which protect the mice against OSA induced impairment of heart function (ejection fraction). Isolated cardiomyocytes of OSA mice exhibited an increase of reactive oxygen species, decrease Ca2+ transients but increase of pro-arrhythmic activity. Nevertheless, some questions and remarks arise that should be addressed prior publication.

  1. Abstract: What authors consider as pro-arrhythmic activity should be defined and included.

Response: We thank the Reviewer for this helpful comment and have clarified the abstract accordingly. Pro-arrhythmic activity was defined as deviations from diastolic Ca baseline between two stimulated Ca transients.

  1. Introduction and Discussion: It is generally accepted that Ca2+ handling disorders resulting in Ca2+ overload and Ca2+ leaks form sarcoplasmic reticulum may trigger ventricular premature beats. However, it should be taken into consideration that Ca2+ disorders and Ca2+ overload deteriorate cell-to-cell electrical ensured by connexin-43 gap junction channels (Tribulova et al. 2009, Andelova et al. 2021). Such conditions result in impairment and block of conduction promoting occurrence of malignant cardiac arrhythmias. Attention should be payed to this fact.

Response: We thank the Reviewer for this worthful comment. Indeed, we have recently found a decreased connexin-43 expression in patients with SDB (Hegner et al., Heart Rhythm 2021). We have therefore revised the manuscript and included the matter of possibly impaired cell-to-cell communication and have included important work from this field (Tribulova et al., Physiol Res 2008; Andelova et al., Int J Mol Sci 2020; Andelova et al., Biomedicines 2022).

The revised Introduction now reads (page 2, lines 50-52):

Dysregulation of cellular Ca handling has also been shown to impair electrical cell-to-cell communication via connexin-43 gap junction channels, further favoring the occurrence of cardiac arrhythmias [12-15].”.

We have also revised the Discussion that now reads (page 10, lines 339-344):

Besides the triggered pro-arrhythmic activity, dysregulation of cellular Ca homeostasis has also been linked to deterioration of electrical cell-to-cell communication, mediated by connexin-43 gap junction channels [12-15]. This may impair electrical conduction, further promoting the occurrence of cardiac arrhythmias [12-15]. Indeed, our group has recently demonstrated that patients with SDB display a decreased atrial connexin-43 expression that also correlated with the occurrence of postoperative atrial fibrillation [27].”.

  1. Moreover, pro-arrhythmic changes or signalling discovered in the study should be defined more specifically as well as discussed.

Response: We thank the Reviewer for this important comment. Pro-arrhythmic, non-stimulated events were defined as deviations from diastolic Ca baseline between two stimulated Ca transients, as stated in the revised methods (page 4, lines 142-144):

“Recordings were analyzed for the frequency of non-stimulated pro-arrhythmic events, defined as deviations from diastolic Ca baseline between two stimulated transients.”

As detected by the Ca-sensitive dye Fura-2-AM, these non-stimulated Ca release events may deplete the SR Ca content, as indicated by reduced caffeine-induced Ca transient amplitude (Figure 5C+D) and reduced steady-state Ca transient amplitude (Figure 4A+B). This is due to increased diastolic SR Ca leak, which resulted in a reduced post-30s-rest Ca transient amplitude ratio (Figure 5A+B). Importantly, ablation of oxidative CaMKII activation (MMVV knock-in mice) protected from this dysregulation in PTFE-treated mice. Increased ROS and cytosolic Ca levels can increase CaMKII activity, as stated in our revised Discussion (page 9, lines 318-319):

“By phosphorylating RyR2 at S2814, CaMKII increases the diastolic open probability of RyR2, thereby increasing the diastolic SR Ca leak [23,28,41].”.

The removal of increased diastolic Ca from the cytosol via Na/Ca exchanger (NCX) results in a net depolarization, as 3 Na ions are transported inward for extrusion of 1 Ca ion. The resulting current can cause triggered activity, leading to arrhythmias.

To further elaborate on the Reviewer’s comment, we have revised the discussion to include the following sentences (pages 9+10, lines 325-338):

“In particular, we found the Ca transient amplitude after 30 s of paused electrical stimulation to be decreased only in OSA wildtype mice, suggesting that the SR Ca leak is increased [11,22]. Consequently, caffeine-induced Ca transient amplitude was decreased in OSA wildtype mice, indicating depletion of the SR Ca content that is in line with the impaired ejection fraction, as measured by echocardiography. Importantly, ablation of oxidative CaMKII activation (MMVV knock-in) protects from this deleterious Ca dysregulation in PTFE-treated mice.

Elevated cytosolic Ca levels following an increased SR Ca leak further increase the activity of CaMKII, possibly resulting in a detrimental vitious circle [42]. Moreover, the cytosolic Ca overload activates Na/Ca-exchanger, which transports Ca to the extracellular compartment [43]. This is an electrogenic process, and the net resulting current favors the occurrence of delayed afterdepolarizations (DADs) that are important triggers for arrhythmias [11,22,28]. Indeed, we observed an increased frequency of pro-arrhythmic non-stimulated events in wildtype but not in MMVV knock-in OSA mice.”.

  1. What is your explanation in support of pro-arrhythmic activity of cardiomyocytes, which exhibited decreased Ca2+ transients resulting in depressed contraction? Since just opposite, increased transients and diastolic Ca2+ concentration is considered as proarrhythmic.

Response: We thank the Reviewer for this comment. In cardiac pathologies such as heart failure, in which increased CaMKII activation also plays a key role, decreased Ca transients and increased diastolic SR Ca leakage and arrhythmias are also observed (Pabel et al. Basic Res Cardiol 2020; Lebek et al. Science 2023). As stated in greater detail above, CaMKII is then able to increase RyR2 open probability via phosphorylation, which increases diastolic SR Ca leak and decreases SR Ca content. Indeed, we observed reduced caffeine-induced Ca transient amplitude and reduced post-rest potentiation, which are indicative of reduced SR Ca content (Figure 5). For a more detailed explanation, please also see our response above.

In contrast to this, increased beta-adrenergic stress can lead to enhanced L-type Ca channel activity, which leads to an increased and more synchronized Ca influx into the cell and greater release from the SR, in return increasing Ca transients (Zhou et al. Proc Natl Acad Sci U S A 2009). In this manner, beta-adrenergic stimulation may also increase diastolic SR Ca leak or stochastic L-type Ca opening and therefore propagation of arrhythmias (Bers, Nature 2002). In our study, we do not believe beta-adrenergic stress to be the key driver of the observed pro-arrhythmic events, but rather increased CaMKII activity similar to heart failure, possibly via increased ROS (Figure 3), leading to an increased diastolic SR Ca leak and loss in SR Ca content. In turn, less SR Ca is available per stimulated contraction, which is congruent with the reduction in Ca transient amplitude (Figure 4A+B) and also with the decreased left ventricular ejection fraction that we observed in vivo (Figure 2 A+C).

  1. How is SERCA activity in your OSA model? Reduced SERCA activity followed by reduced Ca2+ uptake by sarcoplasmic reticulum and Ca2+ overload may contribute to the contractility depression.

Response: We thank the Reviewer for this important comment. In mouse ventricular cardiomyocytes, as we used for the present study, 92% of Ca removal is accomplished by SERCA, while NCX accounts for 7% and other “slow systems” for the remaining 1% (Bers, Nature 2002). Therefore, the relaxation time of stimulated Ca transients, which can be measured as 70 or 90% return time to baseline (RT70 and RT90), for example, is a surrogate marker for SERCA activity if SR Ca leak is unchanged (Ziolo et al., Circ Res 2005). Here, we report unaltered RT70 and RT90 values in all groups in conditions of increased SR Ca leak, therefore it is possible that SERCA activity might be slightly increased in wildtype PTFE mice to compensate for increased SR Ca leak (Figure 4D+E).

  1. Concern the discussion, maybe less is more when focusing on key findings relevant to the aim of the study.

Response: We thank the Reviewer for this comment. During the revision process, we have also revised the Discussion and paid attention to focus on our key findings.

  1. It would be fine to include perspectives in the context of the future research in chapter Conclusions.

Response: We appreciate the Reviewer’s effort to improve the quality of our manuscript. We have therefore added the following perspectives to the revised Conclusions (page 11, lines 393-403):

Indeed, a recent study has demonstrated that CRISPR-Cas9 gene editing is able to specifically rewrite the CaMKIIδ gene in adult mice and to exchange both oxidation-sensitive methionines 281 and 282 with valines, which corresponds to the MMVV knock-in mouse model that we have used in our present study [57]. According to the findings of our study, selective CaMKIIδ gene editing may be both applicable and beneficial also for SDB, which will be investigated in the future. Moreover, detailed analysis of in vivo arrhythmias via implantable transmitters may be an interesting topic for a further study. Besides that, differential metabolism of cardiac energy substrates has been shown to be involved in both cardiac dysfunction and SDB [58-60]. Future studies will aim to investigate this mechanism using our SDB mouse model.”.

Round 2

Reviewer 1 Report

Authors have responded well to my comments. I recommend this manuscript to be published in this journal.